# See More for Scene: Pairwise Consistency Learning for Scene Classification

**Gongwei Chen**[1,2], **Xinhang Song**[1,2], **Bohan Wang**[1,2], **and Shuqiang Jiang**[1,2,3]

[1]Institute of Computing Technology, Chinese Academy of Sciences
[2]University of Chinese Academy of Sciences, Beijing
[3]Institute of Intelligent Computing Technology, Suzhou, CAS
`{gongwei.chen, xinhang.song, bohan.wang}@vipl.ict.ac.cn`
`sqjiang@ict.ac.cn`

## Abstract

Scene classification is a valuable classification subtask and has its own characteristics which still needs more in-depth studies. Basically, scene characteristics are distributed over the whole image, which cause the need of "seeing" comprehensive and informative regions. Previous works mainly focus on region discovery and aggregation, while rarely involves the inherent properties of CNN along with its potential ability to satisfy the requirements of scene classification. In this paper, we propose to understand scene images and the scene classification CNN models in terms of the *focus area*. From this new perspective, we find that large focus area is preferred in scene classification CNN models as a consequence of learning scene characteristics. Meanwhile, the analysis about existing training schemes helps us to understand the effects of focus area, and also raises the question about optimal training method for scene classification. Pursuing the better usage of scene characteristics, we propose a new learning scheme with a tailored loss in the goal of activating larger focus area on scene images. Since the supervision of the target regions to be enlarged is usually lacked, our alternative learning scheme is to erase already activated area, and allow the CNN models to activate more area during training. The proposed scheme is implemented by keeping the pairwise consistency between the output of the erased image and its original one. In particular, a tailored loss is proposed to keep such pairwise consistency by leveraging category-relevance information. Experiments on Places365 show the significant improvements of our method with various CNNs. Our method shows an inferior result on the object-centric dataset, ImageNet, which experimentally indicates that it captures the unique characteristics of scenes.

## 1 Introduction

Image classification is one of the fundamental tasks of computer vision, and attracts a lot of attention as a popular task for facilitating the improvement of convolution neural network (CNN) [16, 23, 26, 11, 14]. As a valuable classification subtask, scene classification has its own characteristics which still needs further study. In comparison with objects, scenes are more complex and have many differences. One of main differences is that the characteristics of scenes are basically distributed over the whole image, while those of objects are confined within a clear boundary. This phenomenon inspires a core idea of scene classification, "seeing" comprehensive and informative regions in the image.

Most scene classification methods aim to extract unspecific [9, 25, 6] or specific regions [29, 33, 3], then aggregate them via statistical models [9, 6, 29] or relation modeling approaches [25, 3]. The main challenge is that the extraction and aggregation are partially independent from the main CNN backbone. It induces some issues as follows. 1) some sort of incompatibility caused by the separated

35th Conference on Neural Information Processing Systems (NeurIPS 2021).

ways to extract and represent regions. 2) inevitable computational consumption from extra region extraction and aggregation processes. It is less studied to investigate the inherent properties of CNN and make them adaptive to the needs of scene classification in a more organic and effective way.

In this paper, we explore a new way to understand scene images and the CNN models for scene classification in terms of the *focus area*. It is defined as the region consisting of pixels with large aggregated activation values in the feature maps. From this perspective, we obtain some observations that reveal the differences in the characteristics of CNNs between scene and object classification, which also correlates well with the properties of these two kinds of images, i.e., scenes are more complex with richer objects. One of the interesting findings is that some properties of training scheme could drive the development of large focus area of the corresponding CNN model, which raises a question about how to optimize the training strategy by considering the scene characteristics.

In order to take advantages of the scene characteristics, we propose a new learning scheme with a tailored loss for scene classification. Our goal is to inspire the CNN models to focus on more regions in the feature maps. However, the supervision of potentially useful regions is difficult to be obtained. In alternative, we attempt to erase the already activated regions (from the images) and require the consistent outputs, thus, the CNN models themselves could expand the focus area with further optimizing. This is also motivated by our preliminary observations that a scene image can retain its semantic meaning even when some regions are masked. Specifically, the modified image is generated by partially erasing the input images with the guidance of model preference. The tailored loss is implemented as pairwise consistency of predictions, which can leverage the category-relevance information.

The principle behind our proposed method is the adversarial learning mechanism under the setting of pairwise consistency. Compared to the original image allows the unconfined exploration, the modified image is used for discovering additional parts which are previously ignored in the original branch. The consistency constraint with the shared parameters unifies these two modes of region explorations, leveraging the underlying opposing operations to perform a more comprehensive and robust information extraction process. In addition, instance-level pairwise consistency shares the excellent quality of knowledge distillation, and yields the unchanged models with superior performances.

Experiments on one of the largest scene datasets, Places365 [38], demonstrate the effectiveness of the proposed method with various CNNs by showing the significant performance improvements. Some statistical analyses about the focus area indicate that our proposed method can enhance the capability of exploring larger focus area in CNNs and fit the characteristics of scene images. We also evaluate our method on ImageNet [15], and the inferior result can be the evidence to reveal the inner working mechanism that is specifically designed for scene images.

## 2   Related work

**Scene classification.** In the era of CNN, many works [9, 5, 7, 18, 3, 31] explore various region extraction and encoding methods based on CNN representations to improve scene classification. They generally need lots of extra overhead compared to a single CNN classifier and are not easy to apply on large scale datasets. A few researches focus on specific deep models designed for scene classification, including Dictionary Learning CNN [19], spatial unstructured layer [10], and contextual relation learning [40]. However, we aim to investigate the inherent properties of scene CNN model, and propose an advanced training method to enhance it without any change of architecture.

**Comparisons of scene and object networks.** A growing number of works have been proposed to understand the discrepancy between these two kinds of networks, especially in terms of the learned representations. In [39], the mean image method is used to visualize the units of CNN layers, and visually exhibits the relevance between the unit structures and the differences of the training data in a qualitative way. Subsequently, a series of works tried to align the units of the CNN with some semantic concepts by measuring the empirical receptive fields [36] or using the feature activation as the segmentation results [1], and found that the networks trained to recognize scenes generally have more interpretable units related to certain concepts, especially objects. In addition, Herranz *et al.* [13] investigated the representations of CNNs trained on ImageNet and Places365 by comparing the object scale ranges in these two datasets, and suggested the transfer performances could be improved when choosing the suitable image sizes. In contrast, we analyze the feature maps by estimating the

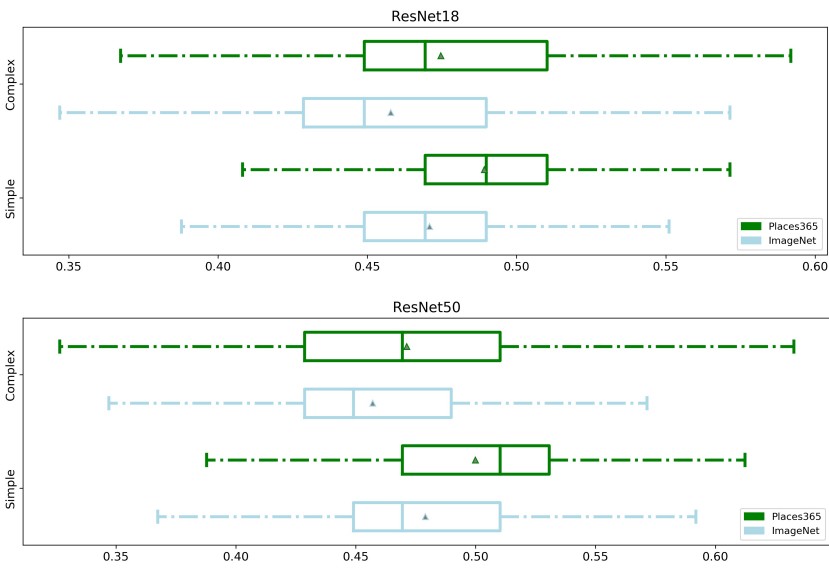

Figure 1: The distribution of the coverage ratio of focus area across various datasets (Places365 and ImageNet) and training schemes (Simple and Complex). The bottom and top of the box show the 25th and 75th percentiles. The center line and triangle in the box denote the median and mean of the data.

aggregated activation values and providing the new perspective of the focus area. Moreover, the training schemes are also introduced and compared in terms of the focus area for better understanding of the difference in the learned CNN models.

**Erasing methods in CNN.** Erasing regions or pixels in images or feature maps is an effective method in weakly supervised object localization [24, 28, 32, 4] / segmentation [17], due to the observation that the models often only concentrate on small and most discriminative regions of object when supervised by only image-level labels. Our idea is inspired by this but study further the properties of CNN from the perspective of the focus area, which changes previous empirical conclusion based on some examples to a robust and reliable statistical analysis. In addition, we explore the erasing method with a tailored loss and extending them into classification task with superior performances, which enriches the understanding of the intrinsic correlations between the erasing method and image classification.

## 3 Comparisons of scene and object networks via the focus area

A large number of previous image classification works propose various technical changes of CNN, and evaluate them on ImageNet [15], which is an object-centric dataset. Despite the proposed large-scale scene-centric datasets, Places205 [39] and Places365 [38], there is still a lack of researches on them, due to the underlying assumption that effective models on object classification could be generalized to scene classification. However, there are always some obvious differences between object and scene images. In this case, scene classification should be investigated individually compared to object classification.

Here, we provide a new perspective to analyze the differences between scene and object networks by estimating the focus area of model on the input images. When the CNN models are trained, it is possible to find out the regions, which the model concentrates on and extracts information from, to make final decision. Various techniques have been developed to achieve this goal by using occluded images [39], gradient information [22], or architecture tweaks [2]. The success of these attempts motivates us to analyze the distributions of focus area of models trained on scene and object datasets, and figure out the possible similarities and differences between them.

Table 1: The classification performance (Top-1 accuracy %) of ImageNet and Places365 with two training schemes, simple and complex ways.

|  | ResNet18 | | ResNet50 | |
|---|---|---|---|---|
|  | Simple | Complex | Simple | Complex |
| ImageNet | 68.80 | 69.93 | 74.72 | 75.78 |
| Places365 | 54.36 | 54.30 | 55.53 | 55.75 |

Based on the feature maps from last convolution layer, we adopt a simple adaptive thresholding method to discover the focus area of model on an input image. The channel-wise average pooling are performed on the feature maps to generate a heatmap. Then the mean value of this heatmap is used as the threshold to segment it into a binary map. Finally, the focus area is composed of the pixels with positive value in the binary map. The underlying assumption is that the feature maps can serve as an indicator to point out the regions which models desired [30, 27], and the pixels with high activation values in the feature maps have more valuable information [1].

The two variables, considered in our analyses about focus area, are dataset and training scheme. In terms of dataset, ImageNet and Places365 are used as object-centric and scene-centric datasets, respectively. Additionally, we introduce two training schemes, including simple and complex ways. The main difference between them is the crop percent in the input processing operations. The former fixes the percent to 76% (224×224 / 256×256), while the latter chooses a random percent in range [8%, 100%] and is widely used in modern CNN training process. The detailed description of these training schemes could be found in appendix. Given a focus area, we compute the coverage ratio of it to the whole image as an analytical metric. As the focus area on a single image may be too specific and not informative, we gather the validation datasets of ImageNet and Places365 to form an evaluation set for analyzing the distribution of the coverage ratio.

We present the classification performance and the visualization of the distribution of the coverage ratio across datasets and training schemes. Referring to the box plots in Figure 1, it is clearly shown that the model learned on Places365 tend to focus more parts of the input image than ImageNet when comparing the median and mean of the data distribution, and the result holds for both training schemes. It is likely that the model trained on Places365 learns the mechanism of comprehensive exploration and activation as a necessary condition for recognizing a scene.

Recalling the definition of the focus area, the pixels in it all have the aggregated activation values than the mean value, which indicates that they either have large activation in a few channels or be frequently activated in many channels. Some evidences [8, 35] have shown that each channel (or unit) in feature maps could be regarded as a part or a full of a detector for a semantic concept, and the concept regions emerge as the pixels with large activation values. In this way, the focus area roughly represents the group of detected semantic concepts in each image, suggesting that the size of the focus area can be accepted as a quantitative index of the regions with semantic concepts. By this correlation, the preference for large focus area seems to be a natural consequence of the scene attributes (containing more semantic concepts).

Another intriguing finding is the effect of training scheme. In contrast with complex training, simple way presents the capability of enhancing the region exploration of model and yielding large focus area on more examples, as shown in Figure 1. We also show the classification performances of two training schemes on ImageNet and Places365 in Table 1. It concludes that simple training is a competitive choice for Places365, while it works worse on ImageNet. We infer that complex training adopts the cropped patches with various scales and yields more diverse training examples, which is better for object classification. However, the similar performances of them on Places365 indicate that the ability of simple training to explore regions is useful for improving the performances, and could bridge the gap caused by the stronger data augmentation. This performance difference also raises question about the optimal training method considering scene characteristics, and motivates us to design a new learning framework as the effective response.

## 4 Advanced training via pairwise consistency

Some earlier works [28, 32, 4] have effectively demonstrated that erasing certain regions in the image or feature maps can force CNN to cover the full extent of object in the context of object classification

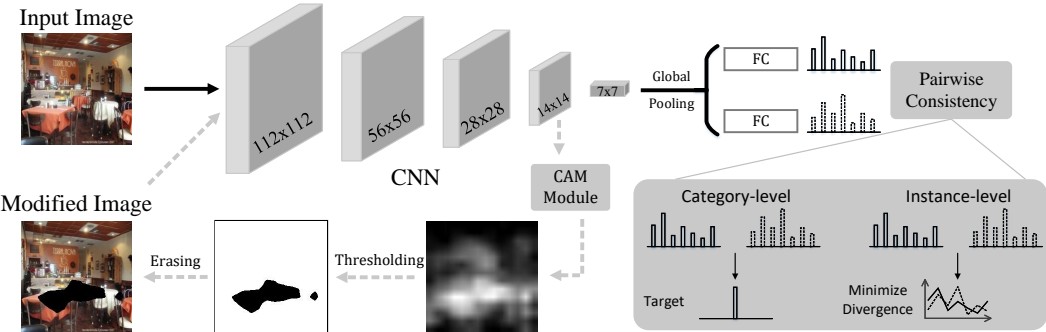

Figure 2: The overview of our proposed method.

along with the classification performance degradation. We conclude that erasing regions can be a useful way to help the CNN models focus on more information of the image, which is reflected by more activated positions in the feature maps.

Based on this, we propose a new learning scheme and design a tailored loss for scene classification. Our design also correlates well with an observation about scene images, which is that if some regions of an image are hiding, we still have a high probability to classify this image as the same category or even the same image. In this way, we decide to make the outputs of an input image and its modified version consistent. The main components of this learning process are the modifying way and the consistency function.

**Modifying image.** Here, we will discuss which regions need to be erased. Normally, It is intuitive to randomly choose regions, which is independent to the model and may not fit well. What is expected is the kind of regions that could reflect the characteristics of images or models and are relevant to the demands of classification.

Fortunately, it is widely recognized [30, 27, 34, 4] that the convolution feature maps can indicate the model preferences in terms of spatial positions. A simple way [30, 4] to leverage this property is to average the feature maps over channel dimensions, which will present the model-relevant regions as a kind of learned knowledge. Furthermore, class activation map (CAM) [37] is proposed to direct the attention to class-specific regions, which reveals the contents that provide evidences for making the corresponding prediction. In our design, class-specific regions (produced by CAM) can present more targeted information, and perform better in deriving region exploration as desired.

The generation of CAM depends on a specific module, termed CAM module, that sequentially contains a convolution layer ($1 \times 1$ in our setting), a global average pooling layer, and a fully connected layer (as a classifier). Then, the output $M$ of CAM is computed as follows,

$$M = \mathcal{F}(A, W^t) = \sum_{c=1}^{C} W_c^t A_c, \tag{1}$$

where $A \in \mathbb{R}^{C \times H \times W}$ is the output from the convolution layer, and $W^t$ is the weights in the classifier to generate the prediction of class $t$. Note, the normalization is performed on the output map as $M = \frac{M - \min(M)}{\max(M) - \min(M)}$. Most modern CNN architectures can meet the demands of CAM generation. So it is quite straightforward to use the original linear classifier of CNN to produce CAM.

However, we attach a CAM module to the input of last down-sampling layer (normally, spatial dimension is $14 \times 14$) for two reasons. One is that larger resolution feature maps may contain more and better detail information. The other is that decoupling the classifier and the generation of CAM would reduce the performance loss caused by erasing more relevant regions derived from the original classifier. In addition, we use the prediction (category with the maximum output from the original classifier) as class $t$, not the ground truth, because the focus point is the model's characteristics, not the properties of the input images.

Guided by the output $M$, we can use a threshold $\gamma$ to convert it to a binary mask $M_B$, which indicates the regions that need to be erased. Then, a modified image $I^*$ is yielded by multiplying the input image $I$ and the binary mask $M_B$. The formulation of this process is shown as follows,

$$I^* = M_B \odot I, \ M_B = T(M, \gamma), \tag{2}$$

where $\odot$ means element-wise multiplication, $T$ denotes the thresholding function.

**Pairwise consistency.** Given the input image $I$ and its modified version $I^*$, we propose a tailored loss function to measure the similarity of their outputs so as to achieve pairwise consistency. Specifically, two consistency forms can be considered, category-level and instance-level. In category-level consistency, we assume that the pair of images belongs to the same category. In instance-level consistency, a stricter constraint is created by enforcing the same prediction of these two images.

In practice, we keep the main component of CNN shared, and only create specific linear classifier for two images, separately. In inference, the classifier of the modified image is discarded and only the original image is used as input to make a prediction. For category-level consistency, two cross-entropy loss with the same ground truth are used. For instance-level consistency, we use Kullback-Leibler divergence as the metric to measure similarity between the output predictions, and also keep supervising the main branch with the ground truth. Our final loss combines these two consistency forms, and is formulated as

$$
\begin{aligned}
L &= l_{main} + l_{side} + \alpha l_{mod} + \beta l_{kl} \\
&= \sum_{k=1}^{K} -g_k \log(p_k) + \sum_{k=1}^{K} -g_k log(r_k) \\
&\quad + \alpha \sum_{k=1}^{K} -g_k \log(q_k) + \beta \sum_{k=1}^{K} -q_k \log(\frac{q_k}{p_k}),
\end{aligned} \tag{3}
$$

where $p, \ q$ denote the predictions of the original and modified images respectively, $g$ is the ground truth (represented as one-hot vector). The item $l_{side}$ indicates the classification loss when applying CAM module and yielding the prediction $r$. We empirically set $\alpha = 1$ and $\beta = 2$ in all experiments.

**Discussion.** The erasing idea is the fundamental element of driving the region exploration of the CNN model. It could be thought as a paradigm that can spawn various specific technical approaches. When applying it on scene classification, a core point is the balance between the loss of important information and the enhancement of exploring regions. Compared with previous alternatively erasing way [24, 28, 32, 4], our pairwise learning method has the advantage of handling these two opposing situations, due to the available complete features in the original image and the drive capability of the erased image. Instance-level pairwise consistency maximizes the benefits by using the category correlation, which sufficiently helps the model expand the focus area and further meets the scene demands. Our method only influences the training stage, and shows no change in inference. Although the proposed method needs double memory usage and computational time during training, we believe it is tolerable as an one-off process.

## 5 Experiments

In this section, we empirically evaluate the effectiveness of the proposed method and conduct some ablation studies for the deeper analyses.

**Datasets. Places365** [38] is one of largest scene-centric datasets with over 10 million labeled images of scenes, which consists of 365 scene categories and two versions of training sets, Places365-challenge and Places365-standard. We choose Places365-standard as the training set in following experiments, which provides 1.8 million images. The validation set has 36,500 images, where 100 images per category. We report Top-1 and Top-5 accuracy (single crop) on the validation set. Meanwhile, for fast evaluation, we also randomly choose 1000 images per category from Places365-standard set, and construct a new training set, termed **Places365-small**. We also reported some experimental results on object dataset, **ImageNet** [15], which contains 1000 object categories with 1.3 million training images and 50,000 validation images.

Table 2: Top-1 and Top-5 accuracy (%) on Places365-small and Places365 dataset. "RF" is short for receptive field, which is computed according to the CNN architectures. "Param." denotes the number of parameters.

| Dataset | Model | Param. | RF | Baseline | | Our Method | |
|---|---|---|---|---|---|---|---|
| | | | | Top-1 | Top-5 | Top-1 | Top-5 |
| Places365 -small | ShuffleNetV2 | 2.85M | 527 | 47.67 | 78.68 | 48.91 | 79.76 |
| | ResNet18 | 11.36M | 435 | 48.21 | 79.22 | 49.70 | 80.62 |
| | ResNet34 | 21.47M | 899 | 48.52 | 79.58 | 50.72 | 81.31 |
| | ResNet50 | 24.26M | 427 | 49.66 | 80.67 | 50.92 | 81.70 |
| | DenseNet121 | 7.33M | 2071 | 49.48 | 80.87 | 51.55 | 82.20 |
| Places365 | ShuffleNetV2 | 2.85M | 527 | 54.01 | 84.05 | 54.95 | 84.56 |
| | ResNet18 | 11.36M | 435 | 54.36 | 84.50 | 55.12 | 84.94 |
| | ResNet34 | 21.47M | 899 | 54.84 | 84.72 | 56.23 | 85.79 |
| | ResNet50 | 24.26M | 427 | 55.53 | 85.85 | 56.61 | 86.15 |
| | DenseNet121 | 7.33M | 2071 | 55.40 | 85.50 | 56.84 | 86.44 |

**Implementation details.** We evaluate the proposed method on several CNN architectures (implemented based on torchvision models[1]), including ResNet [11], ShuffleNetV2 [21], and DenseNet [14]. We use simple data augmentation with random cropping (from $256 \times 256$ to $224 \times 224$) and random horizontal flipping during training. In inference, we apply center-cropping in resized $256 \times 256$ images. All models are trained from scratch by synchronous SGD with momentum 0.9 and a mini-batch size of 256, starting from learning rate 0.1 and decreasing it by cosine annealing strategy [20]. All experiments are conducted with the PyTorch framework on 4 Titan RTX devices. The weight decay is $2e^{-4}$ for ResNet and DenseNet on Places365, $1e^{-4}$ for ShuffleNetV2 on Places365, and $5e^{-4}$ for all models on Places365-small. We conduct experiments on Places365 with 40 epochs and Places365-small with 30 epochs. We simply evaluate the hyper-parameter $\gamma$ in range $\{0.5, 0.7, 0.9\}$, and find $\gamma = 0.7$ is a better choice. Although, we believe more detailed evaluation of $\gamma$ may lead to better performances.

## 5.1 Main results

**Comparison with the baseline.** The main results are shown in Table 2. We compare the baseline and our method with five CNNs (ShuffleNetV2, ResNet18 / 34 / 50, and DenseNet121). It can be observed that the proposed method significantly improves the performances. For example, when applied on ResNet34, the proposed method achieves the largest improvements of 2.20%, 1.39% on Places365-small and Places365 datasets, respectively. Note that these improvements are very significant, because there are less performance differences (e.g. less than 1.5% for ResNet18 and ResNet50) between various models with the standard training scheme.

Interestingly, we also find the positive correlation between the receptive field and the improvement of the proposed method. For example, ResNet34 and DenseNet121 have larger receptive field sizes (899 and 2071) along with bigger relative improvements (4.53% and 4.18%), compared to ResNet50 with smaller receptive field size (427) and relative improvement (2.54%). It suggests that the proposed method benefits much from the large receptive field, as it facilitates the ability of our method to explore more possible useful regions. Another interesting finding is that the performance ranking of ResNet50 and DenseNet121 is different between baseline and our method. Considering the large difference in the receptive field size, we infer that the proposed method is able to leverage the potential value of big receptive field, which may be limited in the standard training scheme.

**Comparison with other erasing methods.** We compare the proposed method with other related erasing methods, HaS [24], ACoL [32], and ADL [4]. All three methods are proposed for weakly supervised object localization task, which concentrate on the characteristics of object images, and suffer degraded classification performances (excluding HaS). While ACoL and ADL achieve the comparable or worse classification performances in WSOL, they clearly outperform the baselines in

---
[1]https://github.com/pytorch/vision

Table 3: Comparison with other erasing methods, HaS [24], ACoL [32], and ADL [4].

| Methods | HaS | ADL | ACoL | Ours |
|---------|-----|-----|------|------|
| ResNet18 | 49.25 | 48.70 | 49.03 | 49.70 |
| ResNet34 | 49.91 | 48.68 | 49.55 | 50.72 |

Table 4: The position of CAM module.

| Model | ResNet18 | ResNet34 | DenseNet |
|-------|----------|----------|----------|
| stage4 | 49.62±0.22 | 49.77±0.08 | 51.29±0.08 |
| stage3 | 49.70±0.10 | 50.72±0.05 | 51.55±0.17 |

Table 5: Comparison with object classification. All experiments are based on ResNet18, and trained with the complex training settings. Top-1 accuracy % (single crop) is reported.

| Dataset | Places365-small | Places365 | ImageNet |
|---------|-----------------|-----------|----------|
| Baseline | 47.80 | 54.30 | 69.93 |
| Ours | 48.68 | 54.78 | 69.57 |

the context of scene classification, as shown in Table 3. Naturally, as a data augmentation method, HaS still show a superior performance when applied on scene images.

The superior results of HaS verify the effectiveness of erasing idea. Compared to it, ADL and ACoL apply the erasing operations on the middle layers, which results in the worse performances. However, all these methods are inferior to our method. This can be attributed to the tailored pairwise consistency loss along with the effective region generation method.

**Comparison with object classification.** We also evaluate the proposed method on ImageNet [15], the most popular object dataset. All comparison experiments in Table 5 use the baseline training procedure in [12] with a total batch size of 256, and 90 epochs, also named complex training scheme in section 3. It can be observed that our method has a negative effect on object classification and show inferior performance compared to the baseline. Note that the complex data augmentation strategy adopt many cropped patches with various scales, which yields more training examples, but also noise especially on scene images. In this way, many small scale patches are hard to be recognized as scenes, leading to the inappropriate situations under the setting of pairwise consistency. Despite this bad impact, the proposed method still obtain better performances. This comparison clearly indicates that our method is specifically designed for scene images and leverage their unique attributes.

## 5.2 Ablation studies

**The position of CAM module.** We evaluate two positions for inserting the CAM module, stage3 (convolution layers operated on $14 \times 14$ feature maps) and stage4 (convolution layers operated on $7 \times 7$ feature maps). Although we assume that the erased image will have the same category distribution as the original input, it may be hard to hold when the erased regions cover lots of discriminative parts. As a result, we insert the CAM module at stage3, which is a little far away from the final classifier and produces the less discriminative regions. As shown in Table 4, it can be seen that this strategy works well on the models with large receptive field, like ResNet34 and DenseNet121, and retain the similar performances on the models with small receptive field, such as ResNet18. Limited by the computation resource, we only report the error bar in Table 4 with 3 repeated runs. The results show that the random deviations are relatively small and do not interfere with our conclusions.

Table 6: Consistent or not? "AM" means Attention Mining loss [17], which aims to minimize the prediction value of the modified image at the ground truth.

| Loss | AM | CPC |
|------|----|----|
| ResNet18 | 48.85 | 49.12 |
| ResNet34 | 48.89 | 49.92 |

Table 7: The effect of two kinds of pairwise consistency and side loss from CAM module. "CPC", "IPC" are short for the category-level and instance-level pairwise consistency, respectively.

| Side | CPC | IPC | ResNet | | |
|------|-----|-----|----|----|----|
| | | | 18 | 34 | 50 |
| ✓ | - | - | 48.75 | 48.76 | 49.52 |
| ✓ | ✓ | - | 49.12 | 49.92 | 50.43 |
| ✓ | - | ✓ | 49.65 | 50.36 | 50.85 |
| ✓ | ✓ | ✓ | 49.70 | 50.72 | 50.92 |
| Baselines | | | 48.21 | 48.52 | 49.66 |

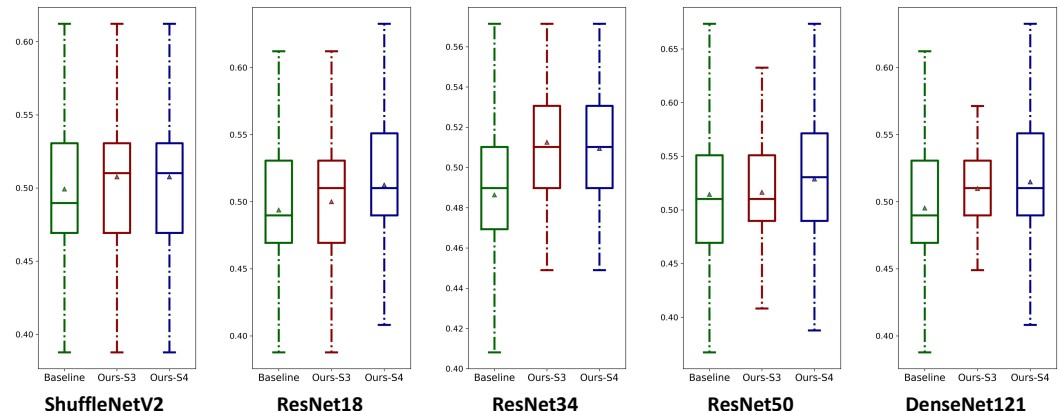

Figure 3: Comparison with the baseline. The focus area is computed with the convolution maps from last convolution layer. "Green", "Red" and "Blue" boxes correspond to the baseline and the proposed method with CAM module inserted into stage3 and stage4 separately.

**Pairwise consistency** is the key factor of the propose method, and motivated by the specific properties of scenes. Firstly, we want to discuss the value of the consistency idea. Li et al. [17] proposed a different loss (attention mining loss) to minimize the prediction value of the modified image at the ground truth. Table 6 shows the results from the attention mining loss and the category-level pairwise consistency loss. We can see that our proposed consistency loss outperforms the attention mining loss, especially on ResNet34. It infers that the consistency target provides a stronger driving force for the model, and is a better way to explore more region information for scenes.

Beyond the category-level pairwise consistency (CPC), we also introduce the instance-level pairwise consistency (IPC) as a more powerful tool. As shown in Table 7, It can be observed that IPC significantly outperforms CPC on various CNNs. It is not surprising because IPC provides a clearer and more precise guidance by investigating the category-relevance information from the prediction distribution. Finally, we combine these two losses to produce a slightly better and robust model.

### 5.3 Analyses of the focus area

Figure 3 shows the distribution of coverage ratio of the focus area from our proposed method with CAM module in stage3 (red box) and stage4 (blue box), and the baseline (green box). The evaluation results are based on Places365 validation dataset and the feature maps from last convolution layer of CNNs trained on Places365-small dataset. It is clearly shown that the proposed method could change the distribution of coverage ratio of the focus area, and force the model to have large focus area on more images, especially when CAM module is inserted into stage4. In terms of the mean and median, the proposed method present significantly larger values than the baseline, except ResNet50 with CAM in stage3 that shows a small difference. It can be attributed to the complex influence factors under the training process including the focus area. In other words, the proposed method may need to find a balance of the enhancement of region exploration, the limitation of network architectures, and even the effects of various losses.

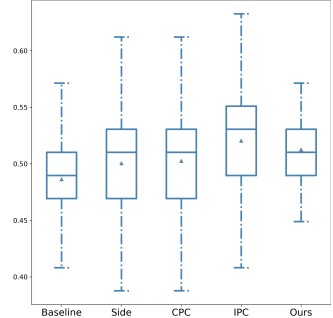

Figure 4: Comparisons of different loss items. The results are evaluated with ResNet34 backbone.

In addition, we also present a detailed comparison of different items in our final loss (described in equation 3) by investigating their effects on the distribution of the focus area in Figure 4. Benefiting from the stronger consistency constraint, instance-level pairwise consistency loss indicates a better driving force of region exploration, compared to category-level pairwise consistency loss. All loss items show the superior results of the focus area to the baseline, which demonstrate the working mechanism of erasing idea

and our pairwise consistency learning method. Unfortunately, there does not seem to be a linear association between the differences in the mean or median of the distribution of different methods and the corresponding classification performances.

# 6 Conclusion

In this work, we propose to investigate the CNN classification model in terms of focus area. Two variables, image datasets and training schemes, is considered when measuring the focus area, and help to analyze the difference between various classification models. An interesting finding is that the focus area in scene networks generally has a bigger size than that in object networks. Meanwhile, the effect of training schemes on the focus area is also significant and inspired us to think the optimal training methods on scene images. Based on this, we propose a new learning scheme and a tailored loss to force CNN to activate more positions in the feature maps for improving scene classification. Extensive experiments on large scale scene dataset, Places365, demonstrate the effectiveness of the proposed method. Meanwhile, the worse performance on object dataset, ImageNet, indicates that our method is specifically designed for scenes and captures their unique attributes. Some comparison analyses also verify the ability of our method to enlarge the focus area.

## Acknowledgements

This work was supported in part by Beijing Natural Science Foundation under Grant L182054 and Z190020, in part by the National Natural Science Foundation of China under Grant 62032022, 61902378 and U1936203, in part by the Lenovo Outstanding Young Scientists Program, in part by the National Postdoctoral Program for Innovative Talents under Grant BX201700255.

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
