# Appendix: See More for Scene: Pairwise Consistency Learning for Scene Classification

## A    More details of the focus area

In this section, we conduct some repeated runs to verify the computational stability of the focus area, and show a detailed analysis of the comparison results under various settings.

**The details of two training schemes.** The simple training way adopts random cropping (from $256 \times 256$ to $224 \times 224$) and random horizontal flipping. The complex training method randomly crops a rectangular region whose aspect ratio is randomly sampled in $[3/4, 4/3]$ and area randomly sampled in $[8\%, 100\%]$, then resizes the cropped region into a 224-by-224 square image and also perform random horizontal flipping on it. All models are trained from scratch by synchronous SGD with momentum 0.9 and a mini-batch size of 256, starting from learning rate 0.1 and decreasing it by cosine annealing strategy (the simple scheme) or by gamma every 30 epochs (the complex scheme). The number of total training epochs and the weight decay are $[40, 2e^{-4}]$ for the simple scheme or $[80, 1e^{-4}]$ for the complex scheme. In inference, we apply center-cropping in the resized $256 \times 256$ images for the simple scheme or the images whose shorter edge is resized to 256 pixels and aspect ratio is kept for the complex scheme.

**The computational stability of the focus area.** In order to investigate the computational stability when computing it based on the single run of the training settings, we perform 3 repeat runs of models trained with simplex and complex schemes, and show mean (the line) and standard deviation (the vertical bar) of these runs in Figure 1. It can be seen that the standard deviations are relatively small and will not change the observations based on a single run. The cross-dataset evaluation result (trained on Places365, evaluated on ImageNet as shown in Figure 1(c)) also indicates the generalization ability of the focus area.

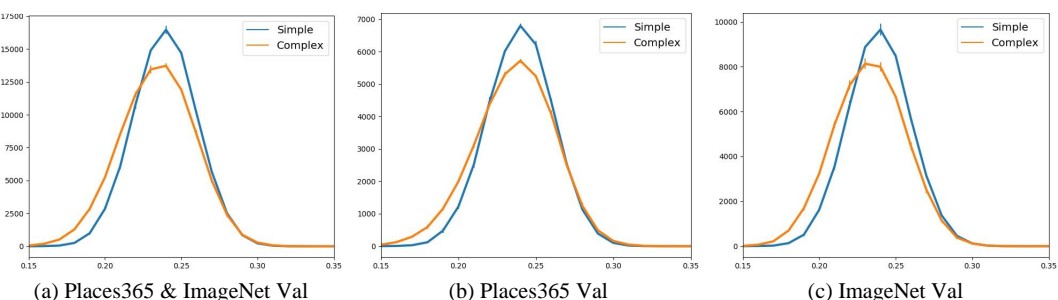

| (a) Places365 & ImageNet Val | (b) Places365 Val | (c) ImageNet Val |

Figure 1: The focus area over 3 repeat runs. The line and vertical bar denote the mean and standard deviation, respectively. "Orange" and "Blue" colors separately correspond to models trained with the simplex and complex training schemes. All the results are implemented with ResNet18 on Places365. Then the focus area is computed on the validation datasets of Places365 and ImageNet.

**Comparison of models trained on Places365.** We also present the focus area of models trained on Places365 dataset in Figure 2. With the proposed training method, all the models show the changed distribution representing more large focus areas in a statistical way and sufficiently capture the rich semantic concepts in scene images. Specifically, ResNet34 and Dense121 yield the better enhancement effects, which may be attributed to the adaptability of the bigger receptive fields to our method.

**More results about different loss items.** Here, we compare the different loss items with two backbones (ResNet18, ResNet34 and ResNet50) in terms of the focus area. The common finding in 3 is that instance-level pairwise consistency has the best effect on driving the model to enlarge the focus area, compared to the baseline and other loss items. However, the combination of two pairwise consistency items generates the best classification performances, suggesting that purely pursuing the maximum effects on expanding the focus area is not an optimal choice for scene classification.

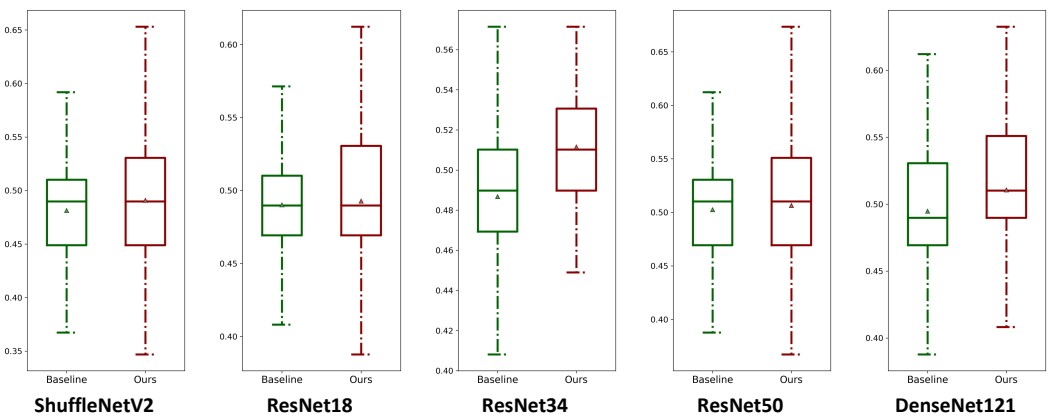

Figure 2: The focus area of models trained on Places365 dataset.

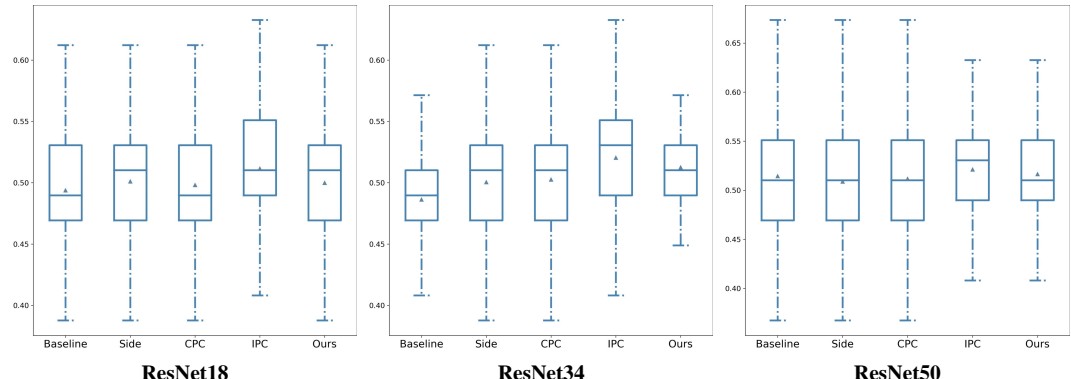

Figure 3: Comparison of different loss items on various backbones. The models are trained on Places365-small dataset.

**Qualitative Analysis.** Figure 4 shows some examples about the changes of the focus area after using the proposed method. The first row presents two groups of success examples that are correctly recognized by our approach, but misclassified by the baseline. The second row indicates two groups of fail examples that are in the opposing condition. In terms of success examples, the "water park" image is classified as "restaurant" by the baseline, while our method expand the focus area by including the water pool and make a correct decision. The similar behavior can be found in the "corridor" image. However, our method can also lead to the reduction of the focus area, like in the "Badlands" image, where the model mainly concentrate on the road so as to yield the prediction "Desert Road". Another possible issue is that even the focus area is expanded, some confusing features (like the stuff in the shelf) are also introduced, resulting in the incorrect result (classifying the "candy store" image as "clothing store").

# B   Experimental configurations of various erasing methods

In section *Experiments* of the main manuscript, the proposed method is compared with various erasing methods that proposed in previous works for different tasks. Here, we will describe their experimental configurations as a supplement.

**Hide-and-seek (HaS)** is a data augmentation method [10], which has been verified effectiveness in various tasks, including image classification. In this method, the key hyper-parameters are drop rate and hide patch size. As this method has been applied in image classification, we following the

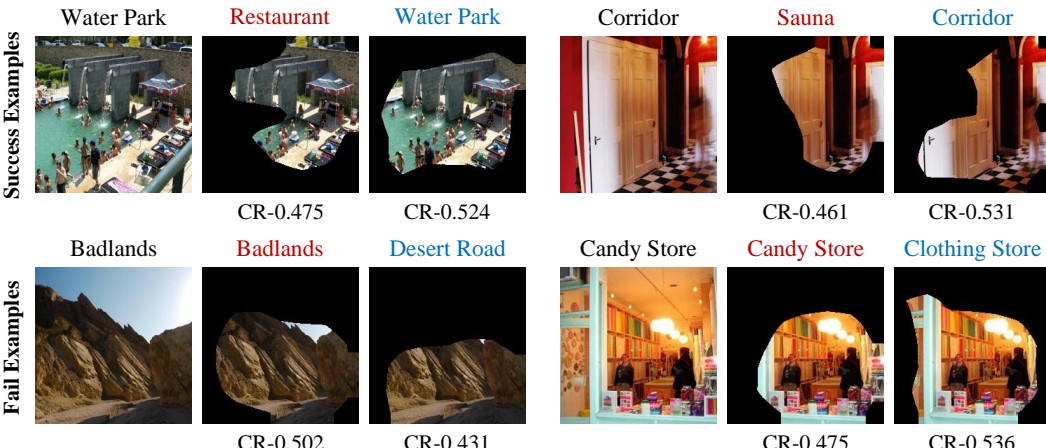

Figure 4: **Qualitative Analysis.** We present four groups of images for analyzing the change of the focus area. The left, middle, and right image in each group denotes the original image and the masked images from the baseline and our method, respectively. The texts above the images represent the ground truth (black), the prediction of baseline (red), and the prediction of our method (blue). "CR" means the coverage ratio of the focus area to the whole image.

setting in the original paper, which is that drop rate equals $0.5$ and patch size is chosen randomly from $\{16, 32, 44, 56\}$.

**Adversarial complementary learning (ACoL)** [11] proposed a two-head architecture where one erases the high-scoring activation positions in the other. We repeat the stage4 as another erasing branch. The erasing threshold is a hyper-parameter, which is set as $0.7$. We report the fusion results from the outputs of two parallel branches.

**Attention-based dropout layer (ADL)** [3] proposed a dropout module that can be applied in various positions of the CNN architecture. We tried the setting (apply ADL in stage3 and stage4) in the original paper, which produces worse results compared to the baseline. It can be concluded that this method has a strong regularization effect on not only object classification, but also scene classification. At last, we only apply it in stage4 and set drop rate and threshold as $0.25$ and $0.8$ respectively.

**Attention mining loss (AMLoss)** was proposed in [7] for weakly supervised object segmentation. It aims to minimize the prediction score of erased image for the ground truth, which can be formulated as follow,

$$L_{am} = P_g(I^*),$$

where $P_g(\cdot)$ means the prediction score for the ground truth class $g$, and $I^*$ denotes the erased image. When combining it with the classification cross entropy loss, we choose the loss weight $\alpha = 1.0$, as other values $(0.5, 2.0)$ have similar performances.

## C   Additional Experiments

Here, we present additional experimental analyses of the CAM module in the proposed method. Meanwhile, the potential negative societal impacts of our method will be discussed in detail.

**Class $t$ in CAM.** In the CAM module, we use the predicted result from the original image as target class $t$ for focusing on the model's characteristics, which is considered better than the ground truth. Here, we present some comparison results in Table 1. It experimentally demonstrates that using prediction has a better impact than the ground truth.

**Societal impacts.** The proposed method does not have direct societal impacts. However, it inherits the implications of CNNs and image classification. For example, it may have the safety risk caused by potential adversarial attacks or the privacy risk. Meanwhile, it may cause severe consequences depending on the application (like autonomous system).

Table 1: The class $t$ in the CAM module. "GT" and "Pred" denote the ground truth and prediction separately.

| Dataset | GT | Pred |
|---|---|---|
| ResNet18 | 49.52±0.12 | 49.70±0.10 |
| ResNet34 | 50.37±0.14 | 50.72±0.05 |

# D  An alternative attempt: Effective Receptive Field

We have made an alternative attempt to analyze the difference between scene and object networks, and the inner working mechanism of the proposed method in terms of effective receptive field (ERF). ERF can be used to reflect the correlation between current pixel and its neighbor pixels, and a large ERF enable the interaction between pixels far apart. When comparing the size of ERF from different networks, we found that scene networks can produce the larger ERF than object networks in the same architecture. We conclude the positive correlation between large ERF and learning scene characteristics, as the useful concepts in a scene image are distributed over the whole space, it would be better to model the dependence of regions far apart for efficiently identifying these visual concepts.

Next, we try to investigate the proposed pairwise consistency learning method in the perspective of ERF. Some comparison results (Figure 7) support the conclusion that our approach could enlarge ERF so as to fit scene characteristics and improve the classification performance. To further study, various losses are evaluated to identify their specific effects on ERF. The proposed tailored loss, instance-level pairwise consistency (IPC), show its superiority ability of enlarging ERF to another variant, category-level pairwise consistency loss (CPC). However, there is no significant difference in the size of ERF among IPC, the side loss, and our final model. When we disable the side loss by inserting CAM module into stage4, it shows the inconsistent changes of ERF (enlarged for three CNNs, reduced for other two CNNs). In conclusion, we believe the main effect of enlarging ERF is from the side loss (the auxiliary classification loss when inserting CAM module into stage3), not the erasing idea or pairwise consistency loss. We think there exists the complex relationship between the proposed method and ERF, which may be caused by multiple non-linear function in the CNN model. Below, we would like to present the analyses about ERF in detail.

## D.1  Backgrounds

ERF is the effective area of a receptive field, which consists of the important pixels that have strong impact on the target output. In an earlier work, Luo et al. [8] theoretically prove that in some cases the distribution of impact within a receptive field distributes as a Gaussian. For that reason, they directly treat effective receptive field as a measurable shape, which size is represented by one or two standard deviation. Here, we present a more general way to measure ERF, and analyze the difference among ERFs derived from CNN models with the same architecture but trained on various datasets.



Figure 5: The impact maps from models without training "INIT", trained on ImageNet "IN", and Places365 "P365".

Following the setting in [8], we measure the impact by the partial derivative $\partial y_{m,n}^l / \partial \mathbf{x}_{i,j}$, where $y_{m,n}$ means the mean value of feature vector indexed by $(m, n)$ in the output feature maps $Y$ of layer $l$, and $\mathbf{x}_{ij}^l$ denotes the pixel vector at position $(i, j)$ of input image $X$. For generating the value of pixel $x_{i,j}$ in the impact map of input, we average the absolute values of gradient vector $\mathbf{x}_{i,j}^l$. All computation can be easily implemented with current deep learning platform.

In practice, we choose the final convolution layer from last stage as layer $l$. When aggregating the features over the channel dimension, due to the weight sharing property in the convolution, all pixels in the aggregated feature map will have the same theoretical distribution of impact. So, we only consider the center point in the output feature maps as $(m, n)$ for simplifying analysis. In order to eliminate the effect of input image, the impact map is presented in terms of

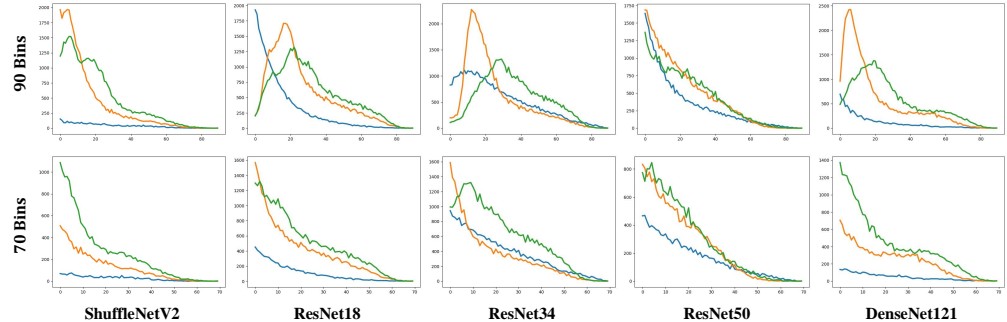

Figure 6: The histograms of the impact maps. Bigger integration area under the lines means larger effective receptive field. "Green", "Orange", and "Blue" denote the results from models trained on Places, ImageNet, and only initialized, respectively.

expectations over input distribution, by averaging the results of one thousand randomly picked images. Figure 5 shows some examples of the impact map.

Modern CNN architectures contain many convolution layers and non-linear layers stacked in an alternating way, which violates the basic assumptions that support the Gaussian distribution conclusion. Since there is no strict theoretical distribution, it is inappropriate to measure ERF with some shape priors (like rectangular or ellipse). We propose to compute a histogram $H$ on the impact map $\mathcal{I}$,

$$H_s = \sum_{i=0}^{HW} \mathbf{I}[\mathcal{I}_i \in r_s], \ s \in \{0, \ldots, S\} \tag{1}$$

where $r_s$ is the $s$-th bin, $S$ is the number of bins, and $\mathbf{I}[\cdot]$ is the indicator function. Before computing histogram, $\mathcal{I}$ is normalized into range $[0, 1]$ by performing min-max normalization, like $\mathcal{I} = \frac{\mathcal{I} - \min(\mathcal{I})}{\max(\mathcal{I}) - \min(\mathcal{I})}$. Using histogram to measure ERF has two main advantages. One is that there is no shape prior as we directly show the distribution of impact. The other is that it may reduce the effects of random noise because calculating depends on a certain range instead of a fixed threshold [8].

### D.2 Comparisons of scene and object networks

As shown in Figure 6, we investigate five kinds of models (ShuffleNetV2_1.5 [9], ResNet18 / 34 / 50 [4], DenseNet121 [5]) trained on two datasets (ImageNet [6] and Places365 [12]). The number of bins is set to 100. We show two kinds of histogram, "90 Bins" and "70 Bins" (remove last 10% and 30% bins with the lowest values, respectively). The effective receptive field size can be calculated as the summation of the numbers within a certain set of bins.

Some observations are listed as follows: 1) ERF is quite small at the initialization, but increases a lot at the end of training; 2) generally, ERF of the model trained on scene images (Places365) is larger than that on object images (ImageNet); 3) when the output feature dimension is large (like 2048 in ResNet50), the difference in ERF between these two datasets becomes small. Inspired by these observations, we try to explore the correlation between large ERF and scene classification.

The core factor in building this correlation is the specific needs ("seeing" comprehensive and informative visual concepts) for classifying scene images. For example, given the categories "office", "home_office", and "living_room", when distinguishing the first two categories, some visual concepts (like shelf) about furniture will be the key elements, but when distinguishing the last two, the visual concepts (like computer, file folder) about working should be paid more attention. Overall, the more categories to classify, the more visual concepts to discover. Based on this characteristic, it infers that "seeing" (classifying) more valuable and informative regions that represent visual concepts is very helpful for scene classification. This conclusion correlates well with the findings in [1], where more object detectors emerge from CNNs trained on scene datasets.

Meanwhile, the fundamental requirement of "seeing" more visual concepts in a scene image is to clarify the relationships between the regions that represent concepts. For example, TV is sometimes very similar to computer monitor, while it may need some references (like sofa) to help identify. This poses a challenge of modeling the dependence of regions or pixels in the image. Furthermore, as the useful concepts in a scene image are distributed over the whole space, it would be better to model the dependence of regions or pixels far apart.

By definition, ERF means the impact of input pixels on the output. Considering the explicit correspondence between the input and output positions, ERF also indicates the correlation betwen target pixels and others, and large ERF enable the interaction between pixels far apart. In this way, it infers that large ERF is yielded in the model trained on scene datasets due to the needs of identifying more semantic regions.

### D.3 Connecting ERF and the convolution feature maps

With measuring impact by the derivative, we note that the effective receptive field depends on the network and the input. If we analyze it over the expectation of input distribution, the focus point will be narrow down to the network (including the architecture and weights). Roughly, there are linear layers (convolution layer, batch normalization layer) and non-linear layers (ReLU, tanh, sigmoid) in CNN[1], which makes it very hard to figure out what changes will be better for large ERF.

To simply analysis, we only focus on the non-linear layer, and analyze the gradient function of ReLU as it is the widest used non-linear operation. The formula of ReLU is $\sigma(x_c) = \max(0, x_c)$, where $c$ means the index of channels. It is easy to deduce the gradient function as follows,

$$\sigma'(x_c) = \begin{cases} 1, & x_c > 0 \\ 0, & x_c <= 0 \end{cases}.$$

(2)

It can be concluded that it is necessary to keep the output of ReLU layer positive for making effective connection. From the perspective of expectation, we can obtain $\mathbf{E}\left[\sum_c \sigma'(x_c)\right] = \sum_c P(x_c > 0)$, with the assumption that $x_c$ has independent distribution along the dimension $C$. This inspires that if we expect larger impact (expectation of summation of gradients over channel dimensions) of input pixels on the output, it is a solution to activate more positions in the output feature maps (keep them positive).

This connection is not easy to build on a single image, due to the complex effects from other layers and their weights. But we may have a relatively clear view when verifying it with a large image set. It has been empirically found [2] that the sizes of activated regions (consisting of spatial positions with positive values in feature maps) from models trained on scene images are generally larger than that on object images, which is consistent with our observation of ERF.

As mentioned in the paper, the erasing idea could drive the model to explore more regions and activate more pixels in the feature maps. Thus, we assume the proposed method could improve the classification performance by enlarging ERF and meet the scene demands. In the following, some comparison results of ERF are introduced for understanding the proposed method.

### D.4 Investigation of ERF with pairwise consistency learning

Figure 7 shows the histograms of the impact maps from our proposed method (orange line) and the baseline (blue line). It is clearly shown that the proposed method could enlarge the effective area of receptive field as desired, compared to the baseline, in terms of the results computed at stage3. For a further study, we compare the changes of ERF of the models trained with different losses in Figure 8. When comparing the proposed variant of pairwise consistency loss, instance-level pairwise consistency (IPC) yields a better result of enlarging the ERF, as shown in the left plot of each black box. However, there is no significant difference in the size of ERF between IPC and side loss, and our final loss even produce the slightly worse results. With these comparisons, we have to admit that

---

[1]Theoretically, the output from batch normalization layer will be connected to all inputs, which means its receptive field is the whole spatial locations. However, we empirically find that the receptive field is independent to the batch normalization layer by using the derivative to measure, which is consistent with the results in [8].

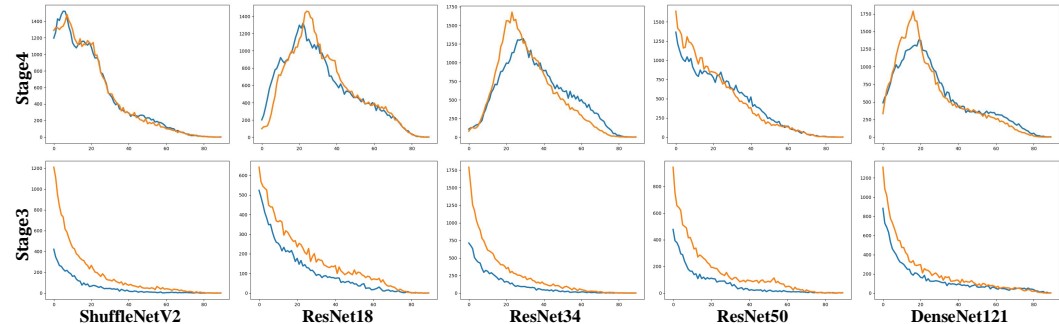

Figure 7: Comparison with the baseline. ERF is computed at the center point of the convolution maps from stage3 or stage4. "Orange" and "Blue" lines correspond to our method and the baseline respectively.

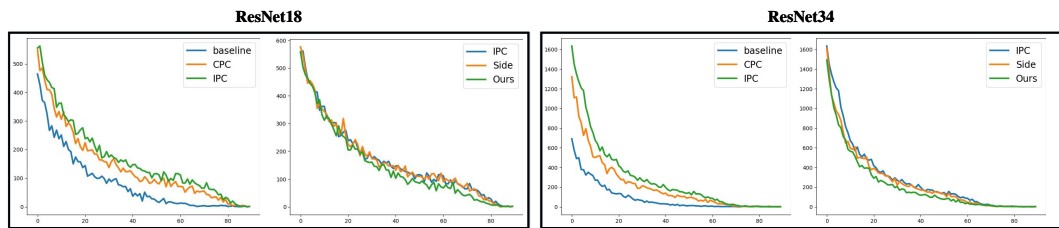

Figure 8: Comparisons of different loss items. ERF is computed at the center point of the convolution maps from the stage3.

the main effect on enlarging ERF is from the side loss (the auxiliary classification loss when CAM module is inserted into stage3), not the erasing idea or pairwise consistency loss.

To verify it, we also compare the baseline and the proposed method with CAM module inserted into stage4, which means no side loss in the final loss function. The comparison results are shown in Figure 9. With three CNNs (ShuffleNetv2, ResNet18, and ResNet34), the proposed approach show the clear evidences of enlarging the ERF as desired. Unfortunately, there is the similar or worse result of the size of ERF, when applying our method with ResNet50 or DenseNet121. To summarize, ERF can be the analytical point to figure out the difference between scene and object networks, but there seems to be more complex relationship between the proposed method and enlarging ERF (the CNN architecture is also an important factor), at least not the consistent positive correlation. In alternative, we introduce the focus area, which could yield the relatively consistent results that make it easy to understand the differences between scene and object networks and the inner working mechanism of our approach.

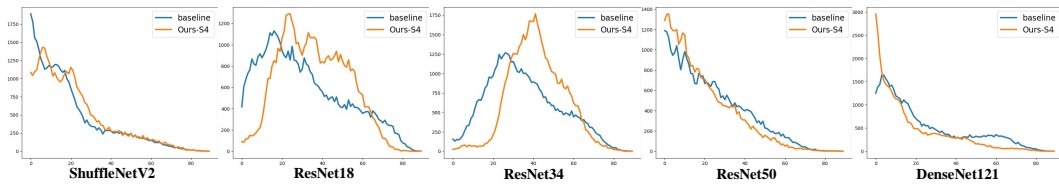

Figure 9: Comparing the baseline and our method with CAM module inserted into stage4 (Ours-S4). ERF is computed at the center point of the convolution maps from the stage4.