# OpenReview forum: "See More for Scene: Pairwise Consistency Learning for Scene Classification"
_NeurIPS.cc/2021/Conference — NeurIPS 2021 Poster_

### Official Review · Reviewer_PqQU · 2021-07-12

**Rating:** 7
**Confidence:** 5

**Summary:**

In this work, the authors enlarge effective receptive fields (ERF) of CNNs for scene image classification. To this goal, the authors propose a new learning scheme and a tailored loss. The proposed learning target is to make the outputs of an input image and its modified version to be consistent. The tailored loss is implemented as pairwise consistency of prediction to leverage category-relevance information. Experiments on Places365 partly verify the effectivenss of the proposed methdods.

**Ethics Review Area:**

["I don’t know"]

**Limitations And Societal Impact:**

The authors have not adequately addressed the limitations and potential negative societal impact of their work.  I suggest the authors add more computation comparisons such as FLOPS/SPEED. Besides, the proposed method should show some failture examples in classification and analysed why the model can predict the true labels under larger ERF.

**Main Review:**

In this work, the authors should address the following concerns.

1.Contributions

In fact, the conceptions of ERF and pairwise learning are not new in current CV field, especically in image classification. Section 3 only list the existing works on ERF and most of the theories are from existing works. The authors only use Histogram to compute the distribution information. In  my view, this is not a contribution, but only a computing method. Besides, the proposed pairwise learning is more like a data augmentation method with erasing modules. In fact, the whole training method is very similar to existing weakly-supervised classification or segmentation methods. In the network structure, the CAM and region erasing methods have already been used in previous works. Thus, the techinical contributions are very limited, considering most of the used modules are from existing works. If there are any insigtful contributions, the authors can highlight them and compare with previous works.

2. Motivations

As far as we known, large ERF is helpful for many CV tasks. The authors aims to enlarge the ERF. However, the basic idea is just using existing methods. What are your essential insights? In my view, there are many methods to modify the image to enlarge ERF, for example recurrent cropping, multi-scale, pyramid structure,etc. How to ensure the effectiveness of the proposed histogram computation and erasing design? Besdies, the basic module is still CAM without any changes.

3.Experiments

First, through the whole experiments, I find that there are no comparisons with recent scene classification methods. In fact, the authors only compare with basic methods with different backbone networks as shown in Tab.1. The performance is not comparable to current SOTA results. Second, there are no full analysis on the key modules. For example, how about using ResNet-101/152 backbone? Why very different ERFs (435 vs 899) show very similar results in Tab.2? If the claims are correct, is it consistent to the experiments? Besides, there are no significant differences in performance of Tab. 6, considering the CPC and IPC. Third, there are no visual comparisions to verify the proposed modules, especially CPC, IPC, Erasing effects, etc. In overall, experiments are not very convincing for the claims.

**Time Spent Reviewing:**

5

---

> ### Author Response · Authors · 2021-08-08
> **Response to Reviewer PqQU (Part 1)**
>
> Thank you for your thoughtful comments.
>
> **Q1: Contributions**
>
> **"In fact, the conceptions of ERF and pairwise learning are not new in current CV field, especically in image classification."**
>
> ERF is originally proposed in NeurIPS 2016 [1], which is introduced in section 3. In this work, by using the definition of ERF, we are the first to investigate the difference in ERF between models under different training condition with the newly proposed metric. In addition, Pairwise learning is a very abstract idea, which could be related to lots of works. For a fair and meaningful comparison, please narrow it down to the specific proposed works (some references). Otherwise, with this rough description, it is unreasonable to judge the novelty or contribution.
>
> **"Section 3 only list the existing works on ERF and most of the theories are from existing works."**
>
> The works related to ERF are very few. We think it is necessary to introduce the essential background about ERF to help the readers understand. Compared to previous works about ERF, we investigate it on models trained on different datasets, which is a new perspective. Some empirical findings are reported, and the correlation between ERF and scene classification is built, which are previously unexplored contents.
>
> **"The authors only use Histogram to compute the distribution information. In my view, this is not a contribution, but only a computing method."**
>
> The historgram is a common method to represent the distribute of data. Its usage in ERF gives some advantages compared to previous methods as discussed in section 3. One is that there is no shape prior, the other is that it may reduce the effects of random noise. We agree the histogram is just a computing method, but using it in ERF is valuable and noteworthy.
>
> **"Besides, the proposed pairwise learning is more like a data augmentation method with erasing modules."**
>
> The main idea behind data augmentation scheme is to enrich the distribution of dataset, so as to improve the model generalization. It is appropriate to regard the erased images as the augmented data. However, the main differences between our method and common data augmentation method is **the data modification way** and **the pairwise consistency**.
>
> the modified image in the proposed approach depends on the current state of the model, while that in data augmentation methods is independent of the model.
>
> In data augmentation, the underlying assumption is that the modified data has the same category with the original one. In the proposed method, when applying instance-level pairwise consistency, the predictions of the pair of images need to be consistent, which is a stronger constraint and provide the useful instance-level categorical relationships.
>
> **"In fact, the whole training method is very similar to existing weakly-supervised classification or segmentation methods. In the network structure, the CAM and region erasing methods have already been used in previous works. Thus, the techinical contributions are very limited, considering most of the used modules are from existing works."**
>
> We have discussed the relevance of existing weakly-supervised classification or segmentation methods in section 2. Compared to these existing works, **we enrich the erasing idea with two valuable points**. One is ERF, which can help us understand the specific effects of the erasing-based training method. The other is the instance-level pairwise consistency, which is a tailored loss for scene classification, represents a more detailed and stronger constraint, and is verified to be effective in experiments (an average improvement of 0.46% in Table 6). Besides these two novel parts, we also try some tweaks (like the position of CAM, the class $t$ in CAM) to provide a more comprehensive exploration of the proposed method.
>
> **"If there are any insigtful contributions, the authors can highlight them and compare with previous works."**
>
> Below, we will summarize the contributions to repsonse to the above comment.
> * We are the first to use ERF to investigate the difference of models trained on different datasets, and find that large ERF is better for scene classification by meeting the specific scene characteristics.
> * The correlation between enlarging ERF and erasing idea is built by the key factor "more activated ReLU units", which enriches the understanding of erasing-based training method.
> * The tailored loss, instance-level pairwise consistency, is proposed to boost the whole training process with a more detailed and stronger constraint.
>
> **Q2: Motivations**
>
> > **Effective Receptive Field & Receptive Field**
>
> * Receptive Field (RF) is a fundamental concept in convolution neural network. It represents the region in the input that connects to a certain output unit.
>
> * Effective Receptive Field (ERF) is originally proposed in NeurIPS 2016 [1]. It is derived from the fact that not all pixels in a receptive field contribute equally to an output unit. The input pixels with large impact on the output can be thought to be more valuable and form the effective area in the receptive field, termed ERF.
>
> **"As far as we known, large ERF is helpful for many CV tasks. The authors aims to enlarge the ERF. However, the basic idea is just using existing methods. What are your essential insights?"**
>
> To the best of our knowledge, there is currently a lack of the exploration of the correlation between ERF and image classification or other CV tasks. It is a consensus that large RF is better for pixel-level prediction tasks, like semantic segmentation [2]. Normally, large RF can lead to large ERF. However, ERF also depends on the network architecture (like non-linear functions, the specific modules), the initialization method, and the training method.
>
> In this work, we explore to enlarge ERF under the same architecture (means RF is not change), which is highly different from the previous works for large RF.
>
> Our basic idea (enlarging ERF for scene classification) is based on the new finding that large ERF is better for scene classification due to the specific scene characteristics (long range interaction between pixels), which is previously unknown.
>
> The essential insights include the difference in ERF between models trained on object and scene images, the correlation between large ERF and scene classification, the connection between ERF and erasing idea.
>
> **"In my view, there are many methods to modify the image to enlarge ERF, for example recurrent cropping, multi-scale, pyramid structure,etc. How to ensure the effectiveness of the proposed histogram computation and erasing design? Besdies, the basic module is still CAM without any changes."**
>
> Recurrent cropping [3] can be regarded as a kind of erasing method with multiple forward process, which is more computationally expensive than the proposed method. Multi-scale and pyramid structure aim to change the network architecture for enriching the features with various scale information. The connection between them and large ERF is unknown.
>
> The proposed histogram metric is used to represent ERF and compare it across various classification models. Based on this metric, we obtain the valuable findings, some of them is consistent with the previous works so as to demonstrate its effectiveness, some of them reveal the important properties of ERF from a new perspective.
>
> The erasing idea is verified to be effective in previous weakly supervised works as introduced in section 4.2. In this work, we enrich this idea by building the correlation between it and enlarging ERF. Also, in addition to previous works, we proposed a new loss, instance-level pairwise consistency, which results in a stronger constraint and superior performances.
>
> ---
>
> [1] Wenjie Luo, Yujia Li, Raquel Urtasun, and Richard Zemel. Understanding the effective receptive field in deep convolutional neural networks. In NeurIPS, pages 4905–4913, 2016.
>
> [2] C. Peng, X. Zhang, G. Yu, G. Luo, and J. Sun, “Large kernel matters - Improve semantic segmentation by global convolutional network,” in CVPR, 2017, pp. 1743–1751.
>
> [3] Yunchao Wei, Jiashi Feng, Xiaodan Liang, and Ming-ming Cheng. Object Region Mining with Adversarial Erasing : A Simple Classification to Object Region Mining with Adversarial. In CVPR, pages 1568–1576, 2017.

---

> ### Author Response · Authors · 2021-08-08
> **Response to Reviewer PqQU (Part 2)**
>
> **Q3: Experiments**
>
> **"First, through the whole experiments, I find that there are no comparisons with recent scene classification methods. In fact, the authors only compare with basic methods with different backbone networks as shown in Tab.1. The performance is not comparable to current SOTA results."**
>
> The main purpose of this work is to discover the unique characteristics of scene classification from the perspective of ERF, and use the erasing-based training method with the tailored loss to improvement scene classification performance by enlarging ERF. For verifing the proposed method, we compare it with not only the standard training method in Table 1, but also **various recently proposed erasing methods** in Table 2, including "HaS", "ADL", and "ACoL". In all experiments, the propose method shows superior performances with the significant improvements.
>
> We believe these comparison and analyses is sufficient to reflect the contribution and value of this work. In fact, there is no recent scene classification method that shares the motivations or technical methods with this work. So, whether to compare with recent scene classification method will have no influence on the main contributions.
>
> Besides, if we only focus on the classification performance, to the best of our knowledge, the current SOTA results on Places365-standard dataset is `58.20%` reported in [1]. However, this result is obtained by combining two resolution inception models, and an unfair comparison for our results. When only comparing the results from the single best model, our result (`56.84%`) is **superior to the current SOTA result** (`56.50%`) in [2].
>
> **"Second, there are no full analysis on the key modules. For example, how about using ResNet-101/152 backbone? Why very different ERFs (435 vs 899) show very similar results in Tab.2? If the claims are correct, is it consistent to the experiments? Besides, there are no significant differences in performance of Tab. 6, considering the CPC and IPC."**
>
> Here, we report the result with ResNet-101 backbone,
>
> our-`56.62%`, baseline-`56.07%`
>
> the improvement is 0.55%, which is significant considering the difficulty of performance improvement with large models. Actually, we believe the proposed method could be generalized on ResNet152 and more other models.
>
> The value in column 4 of Table 1 represents the size of Receptive Field (RF), not Effective Receptive Field (ERF). There are many and complex factors that determine the classification performance. RF is just one of them. Comparing the results of ResNet18/34, it could be seen that the improvement of ResNet34 over ResNet18 is moderate (`0.31%`, `0.48%`) with the standard training scheme, which can be attributed to the overfitting caused by the large amount of parameters (21.47M of ResNet34, 11.36M of ResNet18).
>
> Based on Table 1, we conclude that the proposed method benefits much from the large RF, because of the relatively bigger improvements with ResNet34 and DenseNet121. But considering the results with ResNet101, this conclusion need to be revised as  follows, the proposed method benefits much from the large RF, when the backbone model has an appropriate amount of parameters.
>
> The performance difference between CPC and IPC in Table 6 is `0.46%` on average. It could be described as an siginificant improvement, because of the same training cost. Note the performance difference between ResNet18 and ResNet50 is less than `1.5%`.
>
> **"Third, there are no visual comparisions to verify the proposed modules, especially CPC, IPC, Erasing effects, etc. In overall, experiments are not very convincing for the claims."**
>
> Thank you for pointing it out. As shown in Figure 4, the visual comparisons between the proposed method and the baseline clearly demonstrate that our approach could enlarge the effective area in receptive field as desired, which verifies the effectiveness of the erasing scheme. In addition, as new loss IPC is proposed and experimentally verified to be superior than CPC, it raises a question that if IPC shows a better ability of enlarging ERF compared to CPC. The visual comparisons between IPC and CPC across varisou CNNs demonstrate the superior ability of IPC, and we will add the results and analyses to the appendix.
>
> **Q4: I suggest the authors add more computation comparisons such as FLOPS/SPEED. Besides, the proposed method should show some failture examples in classification and analysed why the model can predict the true labels under larger ERF.**
>
> **R4:** Thank you for pointing this out. Below, we compare the FLOPs of various erasing methods. Note the propose method only effects the training process. So we show Parameters (Param.) and FLOPs of various methods in training and test, respectively.
>
> **Training**
>
> | Method      |    Ours     |     HaS     |     ADL      |     ACoL    |     Baseline    |
> | :-----------  | :----------- | :----------- | :----------- | :----------- | :----------- |
> | Param.      |    11.71M    |    11.36M    |    11.36M    |    22.04M    |    11.36M    |
> | FLOPs      |    3656.84M    |    1821.85M    |    1821.85M    |    2233.46M    |    1821.85M    |
>
> **Test**
>
> | Method      |    Ours     |     HaS     |     ADL      |     ACoL    |     Baseline    |
> | :-----------  | :----------- | :----------- | :----------- | :----------- | :----------- |
> | Param.      |    11.36M    |    11.36M    |    11.36M    |    22.04M    |    11.36M    |
> | FLOPs      |    1821.85M    |    1821.85M    |    1821.85M    |    2233.46M    |    1821.85M    |
>
> It can be seen that the proposed method has twice as much FLOPs as the baseline during training, but the same at test. Consider the training is an one-off process, we think the propose method has no siginificant disadvantage in terms of the computational cost.
>
> About the analysis of failure examples, we will add it to the appendix.
>
> ---
>
> [1] L. Wang, S. Guo, W. Huang, Y. Xiong, and Y. Qiao, “Knowledge Guided Disambiguation for Large-Scale Scene Classification With Multi-Resolution CNNs,” IEEE Trans. Image Process., vol. 26, no. 4, pp. 2055–2068, 2017.
>
> [2] G. Chen, X. Song, H. Zeng, and S. Jiang, “Scene Recognition with Prototype-Agnostic Scene Layout,” IEEE Trans. Image Process., vol. 29, pp. 5877–5888, 2020.

---

### Official Review · Reviewer_BQ7K · 2021-07-14

**Rating:** 7
**Confidence:** 4

**Summary:**

This paper investigated the difference between object classification and scene classification in terms of the effective receptive field. This paper further proposed an erasing scheme guided by the class activation map (CAM) to help enlarge the effective receptive field of classification models during training. Both the original image and the erased image are used to produce the traditional cross-entropy training signal. Also, a KL-divergence term is used to ensure the pairwise consistency between the original image and the erased image. Experiments show minor improvements over baseline methods.

**Limitations And Societal Impact:**

Yes, the authors have adequately addressed the limitations and potential negative societal impact of their work.

**Main Review:**

Originality:
Limited novelty. This paper is the first to investigate the difference between object classification and scene classification in terms of the effective receptive field. But how effective receptive field affects the performance of scene classification is not well studied, neither theoretically nor empirically. The proposed erasing-based training scheme has been used in other lines of research such as multimodal community. Hence the novelty of the proposed approach is limited.

Quality:
Some of the main claims of this paper are simply based on intuition. More theoretical or empirical analysis is required.
1)	About the histograms of impact maps in Figure 2. First to speak, it is confusing to use different numbers of bins. Why is there a difference between 90 bins and 70 bins? Also, some of the charts don’t agree with the claim that ERF of models trained on scene images is larger, e.g. Row 1 Column 3. Lastly, the AUCs of Figure 2 are not consistent, why so?
2)	This paper is based on an intuition that larger ERF will improve the performance of scene classification. But this correlation is well studied. For example, how and to what extent will larger ERF improve performance?
3)	The correlation between the performance gain of the erasing-based training scheme and the enlarged ERF is not studied. This paper merely intuitively provided this chain of causation: erasing scheme during training -> more activated ReLU units -> larger ERF. More comprehensive analysis is required to draw a connection between ERF and scene classification.


Clarity:
This paper is generally easy to follow. However, Section 4.2 about the CAM-guided erasing scheme is quite confusing. How to multiply classification weights with CAM?

Significance:
Moderately significant. The proposed method outperforms various baselines. But the improvements are minor.

Main Weakness:
As mentioned earlier, the erasing-based training scheme is of limited novelty. More analysis is required to draw connections between larger ERF and better scene classification performance.


**Time Spent Reviewing:**

8 hours

---

> ### Author Response · Authors · 2021-08-07
> **Response to Reviewer BQ7K (Part 1)**
>
> Thank you for your thoughtful comments.
>
> **Q1: Limited novelty.**
>
> **"This paper is the first to investigate the difference between object classification and scene classification in terms of the effective receptive field."**
>
> Thank you for the agreement on the originality. Previous scene classification methods mainly focus on the combined features from local and global information, while applying the attributes of scene images into the model design or training has less studied. Meanwhile, Most image classification works are just evaluated on object datasets, leading to the lack of enough testing and analysis about the needs of scene image classification. In this work, we study scene models in a new way, effective receptive field, and reveal the needs of large ERF on scene classification, finally propose the training method with tailored loss for better performance. As a result, we believe this work provides a novel and original perspective on scene classification and shows some useful experiments and analyses.
>
>  **"But how effective receptive field affects the performance of scene classification is not well studied, neither theoretically nor empirically."**
>
> We have made some efforts to understand this effect. The core factor in building the connection between ERF and scene classification is the specific needs ("seeing" comprehensive and informative visual concepts) for classifying scene images.
>
> For example, given the categories "office", "home_office", and "living_room", when distinguishing the first two categories, some visual concepts (like shelf) about furniture will be the key elements, but when distinguishing the last two, the visual concepts (like computer, file folder) about working should be paid more attention. Overall, the more categories to classify, the more visual concepts to discover. Based on this characteristic, it infers that "seeing" (classifying) more valuable and informative regions that represent visual concepts is very helpful for scene classification. This conclusion correlates well with the findings in [1], where more object detectors emerge from CNNs trained on scene datasets.
>
> The fundamental requirement of "seeing" more visual concepts in a scene image is to clarify the relationships between the regions that represent concepts. For example, TV is sometimes very similar to computer monitor, while it may need some references (like sofa) to help identify. This poses a challenge of modeling the dependence of regions or pixels in the image. Furthermore, as the useful concepts in a scene image are distributed over the whole space, it would be better to model the dependence of regions or pixels far apart.
>
> In this work, we leverage a new concept, ERF, as a tool to measure and adjust the dependence of pixels, which requires no need of the architecture change. By definition, ERF means the impact of input pixels on the output. Considering the explicit correpondance between the input and output positions, ERF also indicates the correlation betwen target pixels and others, and large ERF enable the interaction between pixels far apart.
>
> In brief, the dependence of pixels is the fundamental element to achieve "seeing" more concepts so as to meet the needs of scene classification, while ERF can be used to control it.
>
> **"The proposed erasing-based training scheme has been used in other lines of research such as multimodal community."**
>
> The erasing scheme is an abstract concept, and has various implementation methods depending on the problems. We believe it is nescceray to compare and distinguish them in terms of motivation and implementation details. Thus, for clearer and more meaningful comparison, some references in multimodal community is highly expected.
>
> To the best of our knowledge,  the erasing idea is mainly adopted in weakly supervised object localization / segmentation, which has been dicussed in section Related work. In comparison with these related works, **the main differences** can be summarized as two points. 1) We use the pair-wise consistency, especially instance-level pair-wise consistency, to form the erasing-based training framework, which is novel and not explored; 2) The erasing-based training methods is more like an intuitive idea, and previous works use some visulization approaches to analysis and interprete it. While in this work, we connect this kind of idea to effective repceptvie field, and give some interesting findings to ehance the understanding of erasing idea. These findings include earsing image parts can enforce the model to explore more useful regions and increasing the utilization rate of receptive field.
>
> **Q2: About the histograms of impact maps in Figure 2. First to speak, it is confusing to use different numbers of bins. Why is there a difference between 90 bins and 70 bins? Also, some of the charts don’t agree with the claim that ERF of models trained on scene images is larger, e.g. Row 1 Column 3. Lastly, the AUCs of Figure 2 are not consistent, why so?**
>
> **R2:** Thank you for pointing this out. 70 bins is just a paritally enlarged version of 90 bins, which is generated by removing the left 20 points in 90 bins. In the historgrams of impact maps, the rightmost point means the proportion of maximum impact values. As the poistions with large impact value contribute a lot to ERF, we tend to measure ERF based on the distribution of large impact values. So, 70 bins is recommended for comparing ERFs, while 90 bins can depict comprehensive and detailed information including the overall change trend of the impact value.
>
> We conclude that ERF of models trained on scenes is larger than that on objects by verifing it across three kinds of network architectures. Most comparison results support this conclusion. However, the comparison based on ResNet50 exhibits the insignificant difference. We attribute it partially to the bottleneck structure in ResNet50. In the bottleneck module, the number of channels is compressed in Conv3x3 layer for reducing parameters, but may increase the difficulty to enlarge ERF.
>
> the AUCs based on different models (initial, trained on ImageNet and Places) are not consistent, because some bins corresponds smallest impact values are removed when drawing the histograms. As mentioned eariler, the distribution of large impact value is the major factor in measuring ERF. So removing the bins with smallest values has no effect on the observations.
>
> **Q3: This paper is based on an intuition that larger ERF will improve the performance of scene classification. But this correlation is well studied. For example, how and to what extent will larger ERF improve performance?**
>
> **R3:** **"How"** As mentioned eariler, "seeing" more useful visual concepts in scene classification requires the need of modeling long range dependences between parts (regions or pixels) in the image, as the concepts are often distributed over the whole image. ERF is formed by the spatial positions with large impact values, it is easy to infer that large ERF allows current pixel to interact with pixels from far away. In this way, the property of large ERF meets the demand of scene classification, which can be verified by the comparison results between objects and scenes in Figure 2.
>
> To further study this correlation, we raise the question that if there are chances to enlarge ERF for a better satisfaction of the need of scene classification. The proposed method provides a postive answer and demonstrates the correlation by the significant improvements across various CNNs in Table 1.
>
> **"To what extent"** In addition to the above analysis about "how" question, we also thank the review for raising the question "to what extent larger ERF will improve performance". The size of ERF depends on various factors, such as the size of receptive field, the network architecture, the initialization method, and the training method. Thus, it is very complex to analysis the correlation between large ERF and scene classification in general.
>
> In this work, we focus on the effect of the training method on ERF and investigate the performance change of scene classification caused by the enlarged ERF. Unfortunately, the proposed method cannot explicitly control the size of enlarged ERF, which makes it impossible to generate the clear expression of the correlation between the size of ERF and classification performance.
>
> However, there is a relatively rough conclusion that can be provided. With the same architecture, larger RF caused by more stacked 3x3 convolution layers has the bigger potential to achieve the goal of the propose method, considering the difference in performance improvements between ResNet18 (`0.76%`) and ResNet34 (`1.39%`) in Table 1.
>
> ---
>
> [1] D. Bau, B. Zhou, A. Khosla, A. Oliva, and A. Torralba, “Network Dissection: Quantifying Interpretability of Deep Visual Representations,” in CVPR, 2017, pp. 3319–3327.

---

> ### Author Response · Authors · 2021-08-07
> **Response to Reviewer BQ7K (Part 2)**
>
> **Q4: The correlation between the performance gain of the erasing-based training scheme and the enlarged ERF is not studied. This paper merely intuitively provided this chain of causation: erasing scheme during training -> more activated ReLU units -> larger ERF. More comprehensive analysis is required to draw a connection between ERF and scene classification.**
>
> **R4:** The correlation between the performance gain of the erasing-based training scheme and the enlarged ERF can somewhat be obtained with the cross reference on Table 1, Table 2, and Fig. 4.  Fisrt, the gain of using different erasing-based training schemes (see Table 2) can be obtained by comparing with the baseline result in Table 1. Second the gain of enlarging ERF can be obtained by comparing the results in Fig. 4 and Table 4, where the proposed method can enlarge the ERF and obtain better results. The factor "more activated ReLU units" is the core point to connect enlarging ERF and erasing scheme.
>
> In section 4.1, from the definition of ERF and impact value, the gradient function of non-linear layer, ReLU, is analyzed and the connection between large ERF and more activated units is built from the perspective of expectation. Meanwhile, this theory also correlates well with the empirical observation (the size of activated regions in scene models is generally larger than that in object models), as discussed in section *Introduction*. However, this result holds under the condition, the non-linear layer is ReLU. We agree that more general analyses including various non-linear operations could be further studied.
>
> On the other hand, the correlation between more activated units and erasing scheme is empircally verified in previous weakly supervised object localization works. one common finding is that erasing parts in images or feature maps will force the model to explore more useful details and activate more units of feature maps. This empirical finding can be generalized to scene images as it mainly depends on the properties of CNN classification models.
>
> In addition to the chain of "erasing scheme during training -> more activated ReLU units -> larger ERF", as the reviewer suggested, one more important clue is the correlation between ERF and scene recognition. First, our intuition is that scene  images usually consist of more things, requring the CNNs to see and process "more content" in one time. In particular, ERF come somehow measure how large the CNN model can see the content in one time. That's  reason why we seek larger ERF to improve the scene recognition. Second, comparing with the results (between ours and the baseline) in Fig. 4 and Table 1, it can be observed that the larger gain of ERF (see ResNet34 in Fig. 4) can lead to the larger gain in accuracy (see ResNet34 in Table 1), which can also suggest the correlation between ERF and scene recognition performance to some extent.
>
> **Q5: However, Section 4.2 about the CAM-guided erasing scheme is quite confusing. How to multiply classification weights with CAM?**
>
> **R5:** Instead of multiplying classification weights with CAM,  CAM is generated by multiply the weights $W_{t}$ in classifier with the last convolution feature maps $A$.
>
> **Q6: Significance: Moderately significant. The proposed method outperforms various baselines. But the improvements are minor.**
>
> **R6:** We believe our improvements are not that minor. Actually, performance differences in scene classification (especially on Places365) is quite small. For example, the performance difference between ResNet18 (\~70%) and ResNet50 (\~76%) on ImageNet (\~1M images) could be about 6%, while the difference on Places365 (\~1.8M images) is less than `1.5%`. Based on this background knowledge, below we compare the proposed method with baselines in details.
>
> In Table 1, The performance differences between our method with the standard training across 5 Networks on Places365 are `[0.94%,0.76%,1.39%,1.08%,1.44%]`.
> In Table 2, when compared to other erasing methods, the proposed method achieves the improvements of at least `0.45%` with ResNet18, `0.81%` on ResNet34.
> In Table 5, compared to AM, the proposed CPC obtains `0.27%`, `1.03%` improvements with ResNet18/34, respectively.
>
> All above comparison results show significant improvements. Note, in Table 4, there are moderate differences because the popular training setting originally adopted on ImageNet is not suitable for scene classification.

---

### Official Review · Reviewer_DYcV · 2021-07-26

**Rating:** 7
**Confidence:** 5

**Summary:**

To explore the potential recognition ability of CNN for scenes, this paper presents a scene recognition framework based on the analysis of effective receptive field (ERF). This paper first analyzes the correlation between ERF and CNN models of different tasks. Then the authors propose a framework to erase the useful regions while emphasizing other regions and expanding receptive field during training. Experiments are conducted on Places365 and ImageNet. Results show that the proposed method is more effective for scenes than objects.

**Ethical Concerns:**

No ethical issue.

**Limitations And Societal Impact:**

Yes, the authors have discussed the limitations in section 4.2, such as no change in inference and high cost in training.

**Main Review:**

Image classification is a classical and fundamental task in computer vision. Many excellent models have been proposed for object classification (on ImageNet), but the typical researches of scene classification are relatively few. Therefore, it’s interesting to design typical models to exploit the characteristic of scene classification rather than following the same architecture of object recognition.

One interesting insight provided by this paper is that a scene image could be correctly classified even when some regions are covered. And based on this observation, the authors propose a framework to erase the “easier” regions so more other regions could be emphasized during training. Also, the receptive field could be expanded.

Some concerns:
-ImageNet dataset is not introduced.
-As the reviewer claimed in Line 61, “while is not applicable to object classification,” is this an advantage of the proposed method or a disadvantage? And what’s the reason for this.
-In Table 1, which factor is more important to the performance, the number of parameters, or RF size? Also, it seems the gains on Top-1 are larger than Top-5. What’s the reason?
-There are many related results in different tables. Therefore, the differences between them also should be clarified. For instance, ResNet in Table 1 is 54.36, in Table 4 is 54.12. What are the differences between the two settings?

Minor issues:
- Line 30, “as follows.”-> “as follows:”
- It’s better to avoid repeated sentences, such as Line 50 and 177.

**Time Spent Reviewing:**

4

---

> ### Author Response · Authors · 2021-08-04
> **Response to Reviewer DYcV**
>
> Thank you for your positive feedback.
>
> **Q1:** ImageNet dataset is not introduced.
>
> **R1:** Thank you for pointing this out, we will add the necessary introducation. As the most popular object dataset, ImageNet has been the standard benchmark for evaluating image classifcation models. Normally, researchers use a subset that contains 1000 object classes and 1M images for training. In this work, we choose this dataset to evaluate the proposed method, and want to figure out the effectiveness of our method on object classfication. In experiments, the inferior results on ImageNet can indicate that erasing specific parts in images will hurt the classificaiton performance of objects, but help scenes.
>
> **Q2:**  As the authors claimed in Line 61, “while is not applicable to object classification,” is this an advantage of the proposed method or a disadvantage?
>
> **R2:** The advantage of our proposed method is the reflection and effective usage of the unique characteristics of scene classification (in general, erasing some image parts makes no change to the category of scene image). We would like to think this phonemenon that the proposed method is unsuitable for object classification as the evidence for supporting the connection between our method and the characteristics of scene calssification. It indicates the differences in model design of object and scene classification, which also drives us to explore more specific ideas for the scene beyond the existing operations originally designed for the object.
>
>
> **Q3:** In Table 1, which factor is more important to the performance, the number of parameters, or RF size? Also, it seems the gains on Top-1 are larger than Top-5. What’s the reason?
>
> **R3:** In this work, the proposed method can show the consistent improvements under various settings of parameters, RF sizes, and network architectures. Specifically, we empircally find that large RF size has a positive impact on the performance improvements. Compared to the results of ResNet34/50, it is shown that the proposed method enhance the power of large RF, and make a better improvement so as to narrow the performance gap caused by the number of parameters and architectures. In this way, we can conclude that the RF size plays a more important role in our work.
>
> The proposed method forces the model to explore more useful regions in images, which will enhance the model with some subtle but important information and help to distinguish between similar classes. As a result, it shows the better gains on Top1 accuracy than Top5.
>
> **Q4:** There are many related results in different tables. Therefore, the differences between them also should be clarified. For instance, ResNet in Table 1 is 54.36, in Table 4 is 54.12. What are the differences between the two settings?
>
> **R4:** Thank you for pointing this out. For different comparison purposes, some results are reported repeatedly in different Tables, which could be easy to find out. Besides, the difference between results of ResNet18 on Places365 in Table 1 and 4 is due to the training setting.  In this work, the experiments (expect the results reported in Table4) are following the same setting, which is described in section 5. In Table 4, for a fair comparison with the previous results on ImageNet, we follow the popular experimental setting in previous object classification work [1] (which has an advanced data augmentation and some training tweaks).
>
> [1] Tong He, Zhi Zhang, Hang Zhang, Zhongyue Zhang, Junyuan Xie, and Mu Li. Bag of Tricks for Image Classification with Convolutional Neural Networks. In CVPR, pages 558–567, 2019.
>
> **Q5:** Minor issues
>
> **R5:** Thank you for pointing it out, we will revise them.

---

### Decision · Program_Chairs · 2021-09-27

**Decision:**

Accept (Poster)

**Comment:**

This paper presents a systematic study on the potential recognition ability of CNN for scenes in terms of the effective receptive field (ERF).  It is interesting to see that the proposed model is based on exploiting the characteristic of scene classification rather than following the same architecture of object recognition. Moreover, all the reviewers agree that the proposed method provides a key contribution to the fundamental CV tasks.